# Multitarget inhibition of CDK2, EGFR, and tubulin by phenylindole derivatives: Insights from 3D-QSAR, molecular docking, and dynamics for cancer therapy

Khadijah M. Al-Zaydi[1]*, Soukayna Baammi[2], Mohamed Moussaoui[3]*

**1** Department of Chemistry, College of Science, University of Jeddah, Jeddah, Saudi Arabia,
**2** Bioinformatics Laboratory, College of Computing, Mohammed VI Polytechnic University, Ben Guerir, Morocco, **3** Laboratory of Physical Chemistry of Applied Materials, Faculty of Sciences Ben M'Sick, Hassan II University of Casablanca, Morocco

* Kmalzaydi@uj.edu.sa (KMA); moussaouimohamed143@gmail.com (MM)

## Abstract

Cancer remains one of the leading causes of death globally, presenting significant challenges to healthcare systems due to its complexity and the limitations of current therapeutic strategies. Despite advancements in anticancer drug development, monotherapies often fail to provide long-term efficacy due to the emergence of drug resistance. This resistance is primarily due to the activation of compensatory pathways in cancer cells, which allows them to bypass the effects of single-target therapies. To overcome this, targeting multiple key proteins simultaneously has emerged as a promising strategy to enhance therapeutic outcomes and address resistance mechanisms. In this study, 2-Phenylindole derivatives were explored as MCF7 breast cancer cell line inhibitors using 3D-QSAR modeling to design more effective compounds. The CoMSIA/ SEHDA model demonstrated high reliability ($R^2 = 0.967$) and a strong Leave-One-Out cross-validation coefficient ($Q^2 = 0.814$), further validated by external testing ($R^2_{Pred} = 0.722$). Six new compounds with potent inhibitory activity were designed, and their favorable ADMET profiles were confirmed. Molecular docking studies revealed that the newly designed compounds exhibited better binding affinities (−7.2 to −9.8 kcal/mol) to key cancer-related targets (CDK2, EGFR, and Tubulin) compared to the reference drug and the most active molecule (molecule 39) in the dataset. Additionally, 100 ns molecular dynamics simulations confirmed the stability of the best-docked complexes, highlighting their potential as promising candidates for anticancer drug development.

**Data availability statement:** All relevant data are within the manuscript and its Supporting Information files.

**Funding:** The author(s) received no specific funding for this work.

**Competing interests:** The authors have declared that no competing interests exist.

## 1. Introduction

Cancer remains one of the leading causes of death worldwide, posing a significant challenge to healthcare systems due to its complexity and the limitations of current therapeutic strategies [1]. This challenge is further compounded by the fact that, despite advancements in anticancer drug development, monotherapy often fails to achieve long-term efficacy due to the emergence of drug resistance [2]. This resistance arises because a major limitation of single-target therapies is their susceptibility to compensatory pathway activation, which allows cancer cells to bypass drug effects and reduce treatment effectiveness [3]. To address these challenges, targeting multiple key proteins simultaneously has emerged as a promising approach to enhance therapeutic outcomes and overcome resistance mechanisms [4]. Among the most critical molecular targets in cancer therapy are Cyclin-Dependent Kinase 2 (CDK2), Epidermal Growth Factor Receptor (EGFR), and Tubulin, each of which plays a pivotal role in tumor progression and development of drug resistance.

CDK2, a key cell cycle regulator, controls the transition from the G1 to the S phase. Its overactivation leads to unchecked cell division, facilitating rapid tumor growth and contributing to tumor aggressiveness and resistance to apoptosis [5]. EGFR, a receptor tyrosine kinase, is frequently overexpressed or mutated in cancers, activating downstream signaling pathways that promote uncontrolled proliferation, survival, and migration. This dysregulation drives tumor growth, angiogenesis, and metastasis [6]. Tubulin, a structural component of microtubules, is essential for cell division and mitosis. Disruptions in tubulin dynamics can cause chromosomal instability, a hallmark of cancer, and contribute to resistance against microtubule-targeting agents [7]. However, resistance to inhibitors of these proteins often arises due to the development of mutation or activation of alternative survival pathways.

By simultaneously targeting CDK2, EGFR, and Tubulin, this multi-targeted therapy addresses multiple pathways involved in cancer cell survival, proliferation, and metastasis. This approach can potentially prevent or overcome resistance mechanisms that develop with single-target therapies. Moreover, it may enhance treatment efficacy by more effectively controlling tumor growth, reducing the risk of recurrence, and improving patient outcomes. Combining these targets could lead to more durable and potent treatment regimens, particularly for patients with resistant cancers.

The indole nucleus has emerged as a highly versatile scaffold in developing compounds with promising antiproliferative activity, particularly in cancer treatment [8]. A range of 2-phenylindoles has been identified for their ability to inhibit the growth of human breast cancer cells, with the specific mechanisms of action varying depending on the type and position of the substituents on the phenyl ring [9,10]. Notably, recent studies on 2-phenylindole-3-carboxaldehydes have demonstrated their potent antimitotic activity by inhibiting tubulin polymerization, which is crucial for cell division [11]. To address the in-vivo instability of the aldehyde functional group, several modifications, such as the formation of oximes, hydrazones, and other derivatives, were introduced, resulting in compounds with improved stability and continued antimitotic activity [12]. Furthermore, using 3D-QSAR and docking studies, structural optimization of these compounds has provided valuable insights into their interaction with

tubulin, EGFR [13], CDK2 [14] particularly at the inhibitor binding site. These advances have motivated further exploration of novel anticancer agents based on the 2-phenylindole scaffold, focusing on synthesizing stable derivatives with enhanced potency.

Given this background, the present study explores the potential of 2-Phenylindole derivatives as multitarget inhibitors against CDK2, EGFR, and Tubulin. Through an integrated computational approach, including 3D-QSAR modeling, molecular docking, and molecular dynamics simulations, we aim to identify novel compounds with strong and stable binding affinities to all three targets. The proposed compounds may exert a synergistic effect by simultaneously inhibiting these key proteins, disrupting different pathways involved in tumor progression and resistance.

## 2. Materials and methods

### 2.1. Data set

A database of thirty-three compounds was compiled from literature sources [12,15], consisting of novel 2-Phenylindole derivatives, which are being investigated as potential anti-breast cancer agents. The dataset was divided into two groups: twenty-eight compounds constituted the training set, while five compounds, randomly selected, formed the test set to evaluate the model's effectiveness. The chemical structures of the compounds in both the training and test sets are presented in **Fig 1** and **Table 1**. This data was used to develop a 3D-QSAR model and analyze the physicochemical properties of the compounds. For the QSAR analysis, the in vitro biological activity values ($IC_{50}$, in µM) were converted to the corresponding $pIC_{50}$ values ($pIC_{50}$ is the negative logarithm of $IC_{50}$, i.e., $pIC_{50} = 6 - \log10 (IC_{50})$). The 3D structure building and all modeling activities were conducted using the Sybyl 2.0 program package.

### 2.2. Molecular alignment

Molecular structures were first sketched using the sketch module in the SYBYL program and then optimized with the standard Tripos molecular mechanics force field [16] and Gasteiger-Hückel charges [17], using the conjugate gradient method and a gradient convergence criterion of 0.01 kcal/mol. The next crucial step, molecular alignment, was performed to develop an effective 3D-QSAR model. **Fig 2** presents the 3D structure of the core and the superimposed aligned structures of the dataset. The dataset alignment was done using the distill alignment technique in SYBYL [18], with the most active compound 5n, as the template.

### 2.3. CoMSIA analysis

The descriptor fields of the CoMSIA method were computed within a 3D cubic grid with 2 Å dimensions, extending from the edges of the aligned structures in all directions. At each grid point, the steric, electrostatic, hydrophobic,

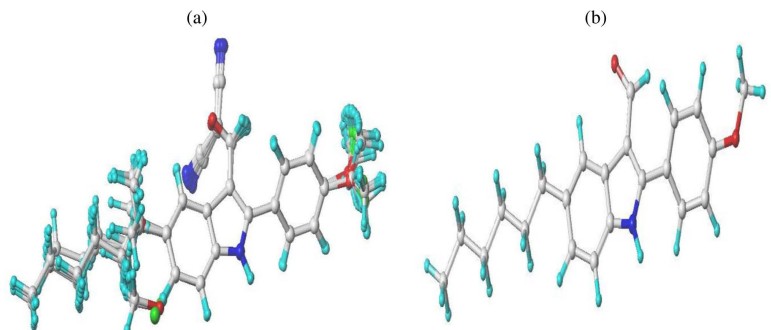

(a)     (b)

**Fig 1. (a) 3D-QSAR structure superposition of training set (b) compound 5n as a template.**

**Table 1. pIC$_{50}$ values of the reported 2-Phenylindole derivatives against EGFR, CDK2 and Tubulin.**

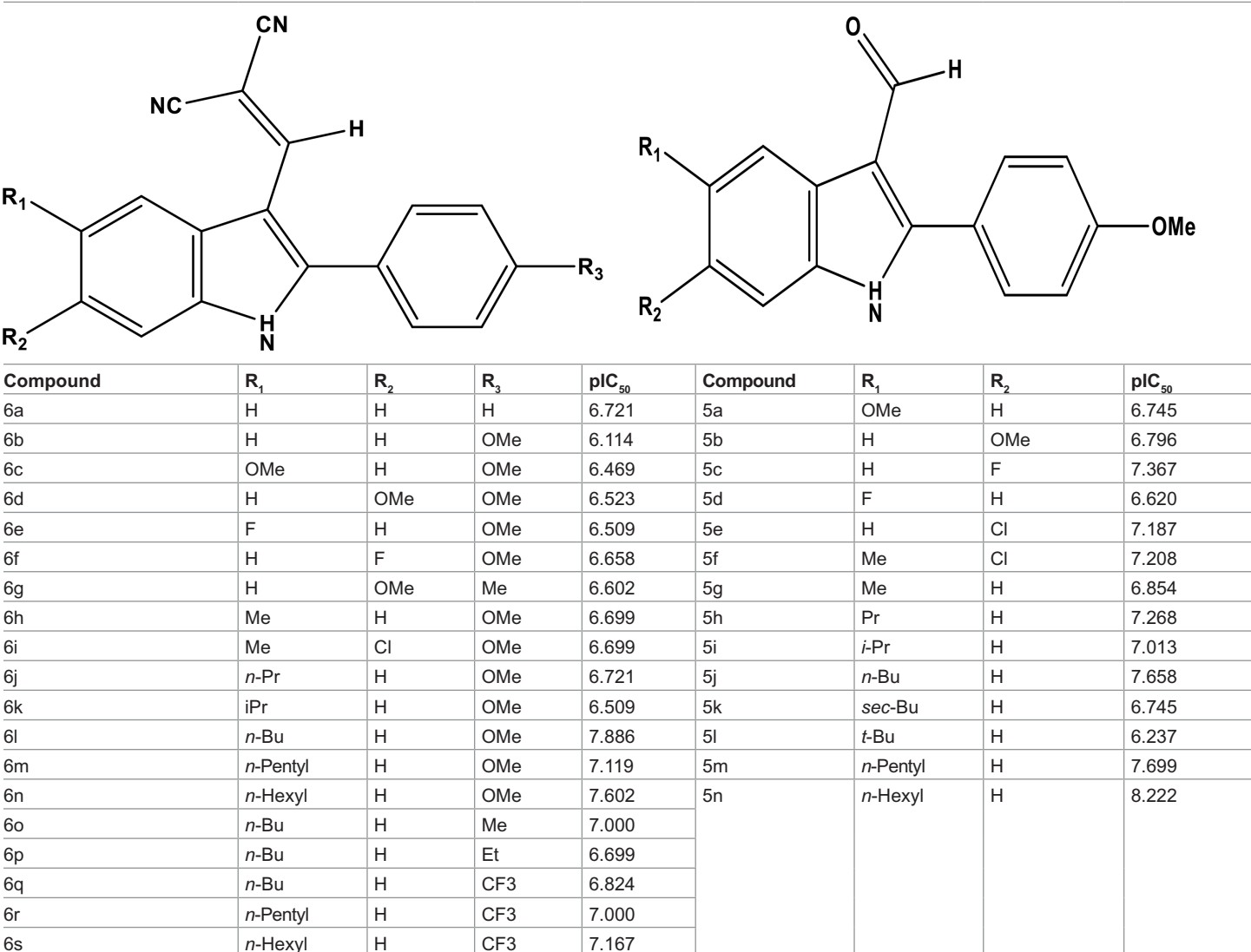

| Compound | R$_1$ | R$_2$ | R$_3$ | pIC$_{50}$ | Compound | R$_1$ | R$_2$ | pIC$_{50}$ |
|---|---|---|---|---|---|---|---|---|
| 6a | H | H | H | 6.721 | 5a | OMe | H | 6.745 |
| 6b | H | H | OMe | 6.114 | 5b | H | OMe | 6.796 |
| 6c | OMe | H | OMe | 6.469 | 5c | H | F | 7.367 |
| 6d | H | OMe | OMe | 6.523 | 5d | F | H | 6.620 |
| 6e | F | H | OMe | 6.509 | 5e | H | Cl | 7.187 |
| 6f | H | F | OMe | 6.658 | 5f | Me | Cl | 7.208 |
| 6g | H | OMe | Me | 6.602 | 5g | Me | H | 6.854 |
| 6h | Me | H | OMe | 6.699 | 5h | Pr | H | 7.268 |
| 6i | Me | Cl | OMe | 6.699 | 5i | *i*-Pr | H | 7.013 |
| 6j | *n*-Pr | H | OMe | 6.721 | 5j | *n*-Bu | H | 7.658 |
| 6k | iPr | H | OMe | 6.509 | 5k | *sec*-Bu | H | 6.745 |
| 6l | *n*-Bu | H | OMe | 7.886 | 5l | *t*-Bu | H | 6.237 |
| 6m | *n*-Pentyl | H | OMe | 7.119 | 5m | *n*-Pentyl | H | 7.699 |
| 6n | *n*-Hexyl | H | OMe | 7.602 | 5n | *n*-Hexyl | H | 8.222 |
| 6o | *n*-Bu | H | Me | 7.000 | | | | |
| 6p | *n*-Bu | H | Et | 6.699 | | | | |
| 6q | *n*-Bu | H | CF3 | 6.824 | | | | |
| 6r | *n*-Pentyl | H | CF3 | 7.000 | | | | |
| 6s | *n*-Hexyl | H | CF3 | 7.167 | | | | |

hydrogen-bond donor, and hydrogen-bond acceptor properties were determined. To quantify these five fields, a probe atom—a charged sp3 hybridized carbon atom with a 1.0 Å radius and a net charge of +1.0—was used at each grid point. The probe atom's slope parameter, which defines the slope of the Gaussian function, was set to its default value of 0.3 [19].

## 2.4. Partial least squares (PLS) analysis

To evaluate the linear correlation between the CoMFA and CoMSIA descriptors and biological activity values, the PLS method [20] was employed. The leave-one-out (LOO) cross-validation method was used in PLS analysis to determine the optimal number of components (N), based on the highest cross-validation correlation coefficient (Q²) and the lowest standard error of estimation (SEE). After identifying the optimal N, non-cross-validated methods were applied to assess the overall significance of the models, using statistical parameters like the coefficient of determination (R²), the F-value (Fisher

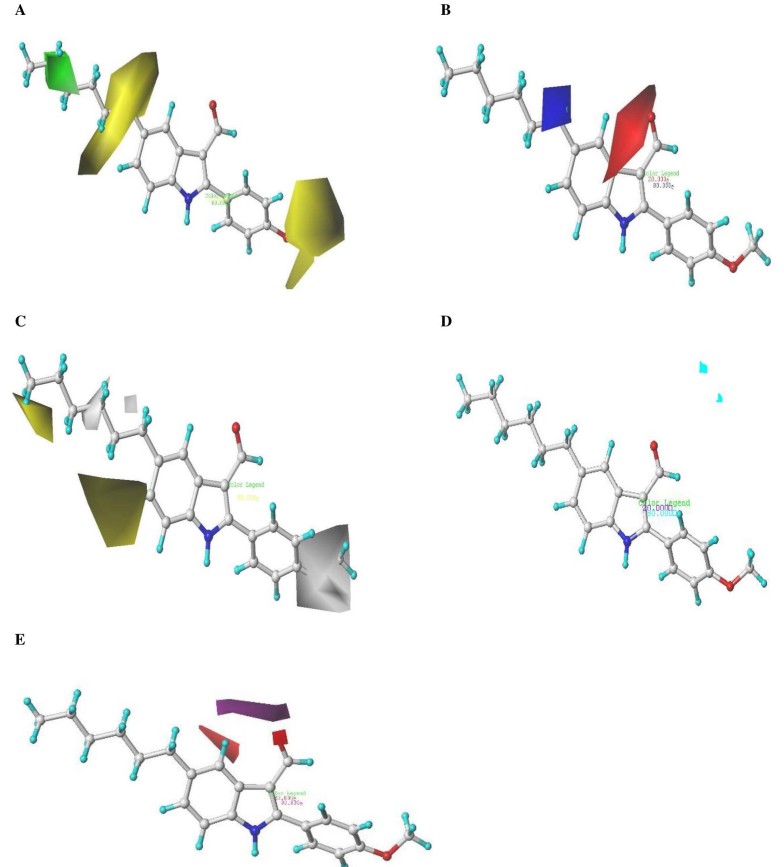

**Fig 2. CoMSIA contour maps: (A) Steric, (B) Electrostatic, (C) Hydrophobic, (D) Hydrogen bond donor, and (E) Hydrogen bond acceptor fields, displayed with a grid spacing of 2.0 Å and combined with compound 5n.**

test), and the standard error of estimation (SEE). Additionally, several external validation strategies were utilized to further evaluate the robustness and statistical validity of the established models [21]. The equation for SEE is presented below:

$$SEE = \sqrt{\frac{PRESS}{n-c-1}}$$

Where n represents the number of compounds, c represents the number of components, and PRESS is the sum of squared deviations between the predicted and actual activity values for each molecule in the test set.

## 2.5. Molecular docking studies

The three-dimensional structures of 3 protein targets, were retrieved from the Protein Data Bank (RCSB) via https://www.rcsb.org/ (**Table 2**). These structures were visualized using UCSF Chimera [22] and prepared with MGLtools (version 1.5.6, The Scripps Research Institute, La Jolla, CA, USA) [22]. Preprocessing involved removing water molecules, heteroatoms (hetatm), and co-crystallized ligands. Subsequently, polar hydrogen atoms were added, Gasteiger charges were assigned, and the structures were converted to pdbqt format for further analysis [23]. The grid box spacing was set to 0.375 Å, centered on the regions where co-crystallized ligands interact with active site residues. Docking simulations were conducted for all 3 protein targets, generating nine poses per protein-ligand complex based on docking affinity. The

**Table 2. The selected targets and the coordinates of the grid box.**

| Protein | PDBID | Grid box center (Å) | Grid box size(Å) |
|---|---|---|---|
| Tubulin | 1SA0 | center_x = 117.219<br>center_y = 90.179<br>center_z = 6.289 | size_x = 20<br>size_y = 18<br>size_z = 40 |
| Epidermal growth factor Receptor (EGFR) | 1M17 | center_x = 21.697<br>center_y = 0.303<br>center_z = 52.093 | size_x = 42<br>size_y = 18<br>size_z = 22 |
| Cyclin-dependent kinase (CDK2) | 2A4L | center_x = 100.865<br>center_y = 101.746<br>center_z = 79.893 | size_x = 40<br>size_y = 40<br>size_z = 40 |

docking outcomes were visualized and analyzed using Discovery Studio Viewer to identify critical interactions between ligands and protein binding sites [24]. For each protein target, the conformation with the lowest binding affinity (as indicated by docking scores) and the highest number of bonds was chosen as the initial binding mode for subsequent molecular dynamics simulations.

## 2.6. Molecular dynamics simulations (MDs)

Molecular dynamics (MD) simulations were performed using GROMACS 2019.3 [25] to evaluate the stability and binding mechanisms of protein-ligand complexes involving the designed and active compounds. The CHARMM27 force field was used to construct protein topologies, while ligand topologies were generated using the SwissParam server [26]. Each complex was positioned within a dodecahedral box (1.0 nm) filled with TIP3P water molecules and neutralized with counter ions [27]. Energy minimization was conducted using the steepest descent algorithm with a maximum force threshold of 1000 kJ/mol/nm [28]. To equilibrate the systems, two consecutive 1 ns simulations were performed under NVT and NPT ensembles at 300 K and 1 bar, regulated by the Berendsen thermostat and Parrinello–Rahman barostat, respectively. Periodic boundary conditions (PBC) were applied throughout the simulations, and long-range electrostatics were calculated using the particle mesh Ewald (PME) method. High-frequency bonds involving hydrogen were constrained using the LINCS algorithm, allowing the use of a 2-fs integration time step [29]. The MD simulations were conducted over 100 ns, totaling 50,000,000 steps, with coordinates recorded every 2 fs. The output trajectories were generated, and the corresponding data files were analyzed to gain a deeper understanding of the protein's behavior.

## 3. Results and discussion

### 3.1. CoMSIA results

To construct a robust 3D-QSAR model, molecule 5n, which exhibits the highest inhibitory activity, was chosen as a reference for data alignment. This alignment was crucial for generating contour maps in both CoMFA and CoMSIA analyses, as illustrated in **Fig 1**.

The primary objective of this phase is to develop reliable CoMFA and CoMSIA models by correlating the observed and predicted $pIC_{50}$ values for the training and test sets. In the CoMFA model, steric and electrostatic fields were integrated. On the other hand, the CoMSIA model was developed using thirty-one different combinations of steric, electrostatic, hydrophobic, hydrogen-bond donor, and hydrogen-bond acceptor fields. The most effective model was identified by evaluating the highest coefficient of determination values for both non-cross-validation ($R^2$) and cross-validation ($Q^2$), alongside the lowest standard error of estimate (SEE), the minimal number of principal components (N), and the most significant F-test value (F). Among the tested models, the SEHDA CoMSIA model provided the most accurate predictions of biological activity, demonstrating the most favorable statistical metrics **Table 3**. This model achieved an $R^2$ of 0.814, utilized six optimal principal components, and had a reliable Standard Error of Estimate (SEE) of 0.091 and an F-test value of

**Table 3. Statistical results of CoMSIA models with different combinations of molecular fields.**

| Generated model | Q² | N | SEE | R² | F | R²pred | Fractions | | | | |
|---|---|---|---|---|---|---|---|---|---|---|---|
| | | | | | | | S | E | H | D | A |
| CoMSIA/S | 0.568 | 2 | 0.185 | 0.864 | 22.325 | 0.642 | 1 | | | | |
| CoMSIA/H | 0.703 | 3 | 0.134 | 0.929 | 45.762 | 0.597 | | | 1 | | |
| CoMSIA/SE | 0.788 | 5 | 0.111 | 0.951 | 68.498 | 0.601 | 0.520 | 0.480 | | | |
| CoMSIA/SHA | 0.702 | 4 | 0.106 | 0.956 | 75.466 | 0.566 | 0.245 | | 0.471 | | 0.284 |
| CoMSIA/HDA | 0.698 | 4 | 0.140 | 0.922 | 26.32 | 0.693 | | | 0.664 | 0 | 0.336 |
| CoMSIA/SEDA | 0.781 | 5 | 0.100 | 0.960 | 84.796 | 0.602 | 0.430 | 0.388 | | 0 | 0.182 |
| CoMSIA/SHDA | 0.702 | 4 | 0.107 | 0.955 | 73.679 | 0.670 | 0.342 | | 0.408 | 0.001 | 0.249 |
| **CoMSIA/SEHDA** | **0.814** | **6** | **0.091** | **0.967** | **102.992** | **0.722** | **0.183** | **0.343** | **0.337** | **0.001** | **0.136** |

R² represents the square of the non-cross-validated coefficient.

Q² is the square of the leave-one-out (LOO) cross-validation coefficient.

R²pred denotes the square of the prediction coefficient.

N refers to the optimal number of components.

SEE stands for the standard error of estimation in non-cross-validated analysis.

F is the value obtained from the F-test.

S, E, H, D, and A correspond to the steric, electrostatic, hydrophobic, hydrogen-bond donor, and hydrogen-bond acceptor contributions, respectively.

102.992. Additionally, **Table 4** summarizes the predicted $pIC_{50}$ values and CoMSIA/SEHDA descriptors for the compounds analyzed.

The results in **Table 5** show that the CoMSIA/SEHDA model has met all evaluation criteria and aligns with the standards set by Golbraikh, and Tropsha [30–32]. This model offers a deeper understanding of activity compared to the CoMFA model, with enhanced predictive capabilities for new compounds and adherence to all required validation protocols. Consequently, we used the CoMSIA/SEHDA contour maps to elucidate the structural elements that enhance activity and to aid in the discovery of new active compounds.

### 3.2. Graphical Analysis of the CoMSIA Model

The CoMSIA contour map visualizes data from the selected 3D-QSAR model, with compound 5n as the reference. **Fig 2** presents the SEHDA model's contour map for steric, electrostatic, hydrophobic, hydrogen bond donor, and hydrogen bond acceptor fields. In the steric contour maps depicted in **Fig 2A**, green contours (80% contribution) indicate areas where bulkier substitutions enhance biological activity, whereas yellow contours (20% contribution) identify areas where such substitutions diminish activity. The electrostatic maps in **Fig 2B** show blue contours (80% contribution) highlighting regions where positive electrostatic groups are beneficial, and red contours (20% contribution) where negatively charged groups are favored. In the hydrophobic maps, illustrated in **Fig 2C**, yellow contours (80% contribution) point out favorable hydrophobic regions, while white contours (20% contribution) indicate advantageous hydrophilic areas. The hydrogen bond donor maps in **Fig 2D** feature cyan contours (80% contribution), suggesting that hydrogen bond donor groups boost activity, and purple contours (20% contribution) indicate less favorable regions. Lastly, in the hydrogen bond acceptor maps shown in **Fig 2E**, magenta contours (80% contribution) mark areas where hydrogen bond acceptor substitutions enhance activity, and red contours (20% contribution) show where they hinder activity.

### 3.3. Design of new drug candidates

The contour map analysis of the CoMSIA models provided a foundation for designing novel inhibitors. Using this analysis, we identified key structural features essential for activity, which guided the optimization process. A summary of the design

Table 4. Predicted pIC$_{50}$ values and corresponding CoMSIA descriptors for the compounds in this study (*: test set).

| Compounds | pIC$_{50}$ | pIC$_{50}$ pred | S | E | H | D | A |
|---|---|---|---|---|---|---|---|
| 5a | 6.745 | 6.696 | 7.92 | 1.20 | 4.45 | 1.42 | 2.34 |
| 5b | 6.796 | 6.901 | 7.92 | 1.19 | 4.50 | 1.42 | 2.34 |
| 5d | 6.620 | 6.633 | 7.48 | 1.08 | 5.21 | 1.42 | 2.34 |
| 5e | 7.187 | 7.184 | 7.50 | 1.06 | 6.11 | 1.42 | 2.96 |
| 5g | 6.854 | 6.998 | 7.83 | 1.10 | 5.48 | 1.42 | 2.34 |
| 5h | 7.268 | 7.272 | 8.51 | 1.11 | 6.11 | 1.42 | 2.33 |
| 5i | 7.013 | 6.931 | 8.58 | 1.09 | 6.12 | 1.43 | 2.33 |
| 5j | 7.658 | 7.569 | 8.84 | 1.10 | 6.44 | 1.42 | 2.34 |
| 5k | 6.745 | 6.816 | 8.92 | 1.10 | 6.42 | 1.42 | 2.34 |
| 5l | 6.237 | 6.262 | 8.99 | 1.24 | 6.55 | 1.42 | 2.34 |
| 5m | 7.699 | 7.704 | 9.13 | 1.11 | 6.72 | 1.42 | 2.34 |
| 5n | 8.222 | 8.067 | 9.42 | 1.10 | 7.02 | 1.42 | 2.34 |
| 6a | 6.721 | 6.720 | 7.60 | 1.45 | 5.91 | 1.42 | 2.78 |
| 6c | 6.469 | 6.474 | 8.44 | 1.61 | 4.94 | 1.42 | 2.8 |
| 6d | 6.523 | 6.558 | 8.43 | 1.60 | 4.97 | 1.42 | 2.79 |
| 6e | 6.509 | 6.417 | 8.03 | 1.55 | 5.64 | 1.42 | 2.77 |
| 6f | 6.658 | 6.651 | 8.03 | 1.55 | 5.63 | 1.42 | 2.77 |
| 6g | 6.602 | 6.475 | 8.34 | 1.53 | 5.93 | 1.42 | 2.77 |
| 6h | 6.699 | 6.681 | 8.35 | 1.53 | 5.89 | 1.42 | 2.77 |
| 6i | 6.699 | 6.668 | 8.36 | 1.53 | 6.95 | 1.42 | 2.74 |
| 6j | 6.721 | 6.792 | 9.00 | 1.54 | 6.47 | 1.43 | 2.74 |
| 6k | 6.509 | 6.493 | 9.06 | 1.53 | 6.46 | 1.42 | 2.76 |
| 6m | 7.119 | 7.163 | 9.59 | 1.53 | 7.07 | 1.42 | 2.78 |
| 6n | 7.602 | 7.668 | 9.88 | 1.53 | 7.36 | 1.42 | 2.77 |
| 6o | 7.000 | 7.158 | 9.23 | 1.44 | 7.54 | 1.42 | 2.78 |
| 6q | 6.824 | 6.697 | 9.26 | 1.70 | 8.42 | 1.42 | 2.77 |
| 6r | 7.000 | 7.010 | 9.55 | 1.71 | 8.65 | 1.42 | 2.77 |
| 6s | 7.167 | 7.208 | 9.83 | 1.71 | 8.88 | 1.42 | 2.76 |
| 5c* | 7.367 | 7.029 | 7.50 | 1.08 | 5.32 | 1.42 | 2.34 |
| 5f* | 7.208 | 7.049 | 7.83 | 1.08 | 6.65 | 1.43 | 2.33 |
| 6b* | 6.114 | 6.677 | 8.02 | 1.54 | 5.58 | 1.42 | 2.77 |
| 6l* | 7.886 | 7.181 | 9.30 | 1.53 | 6.95 | 1.42 | 2.78 |
| 6p* | 6.699 | 7.085 | 9.54 | 1.44 | 7.95 | 1.42 | 2.78 |

process is illustrated in **Fig 3**, with compound 5n, the most active compound in the dataset, selected as the template for further modifications (**Fig 3**). The CoMSIA/SEHDA model, which demonstrated superior external validation, was then employed to predict the pIC$_{50}$ values of the newly designed compounds. This approach successfully identified six candidate compounds with predicted activity values exceeding the reference molecule, compound 5n (pIC$_{50}$ = 8.222). The structures and predicted activity values of these compounds are detailed in **Fig 4**, underscoring their potential as promising inhibitors.

### 3.5. ADME and toxicity profiling

The failure of many drug candidates during clinical development is often attributed to poor blood-brain barrier permeability, toxicity, or insufficient efficacy [33]. Therefore, predicting and optimizing the ADMET (Absorption, Distribution, Metabolism, Excretion, and Toxicity) properties of new chemical compounds is critical to avoid complications in the later stages of

**Table 5. Statistical parameters for validating the CoMSIA/SEHDA model.**

| Statistical parameter | Score | Threshold | Comment |
|---|---|---|---|
| $R^2_{pred}$ | 0.722 | More than 0.600 | Passed |
| $R_0^2$: Determination coefficient for the plot of predicted against observed at zero intercept | 0.882 | More than 0.600 | Passed |
| $R_0'^2$: Determination coefficient of the plot of observed versus predicted at zero intercept. | 0.978 | More than 0.600 | Passed |
| $\left\lvert R_0^2 - R_0'^2 \right\rvert$ | 0.096 | Less than 0.300 | Passed |
| $\frac{R^2 - R_0^2}{R^2}$ | -0.221 | Less than 0.100 | Passed |
| $\frac{R^2 - R_0'^2}{R^2}$ | -0.354 | Less than 0.100 | Passed |
| K: Zero intercept slope of predicted against observed activity for the test set | 1.008 | $0.85 \leq K \leq 1.15$ | Passed |
| K': Zero intercept slope of observed against predicted activity for the test set | 0.987 | $0.85 \leq K' \leq 1.15$ | Passed |

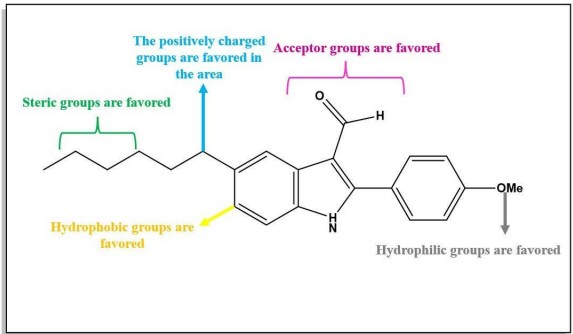

**Fig 3. Structure-activity relationship derived from CoMSIA- SEHDA.**

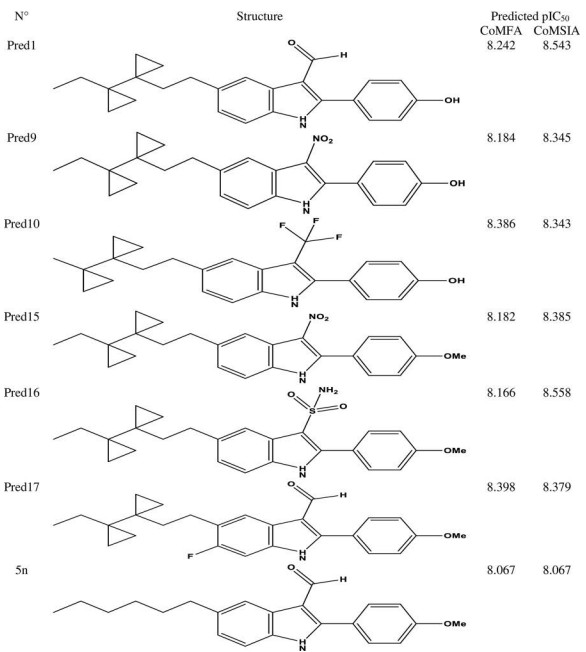

**Fig 4. Structures and Predicted pIC$_{50}$ Activities (Pred) According to the 3D-QSAR Model for Predicted Compounds.**

drug development [34]. This study evaluated the ADMET pharmacokinetic parameters of fifty-nine newly designed compounds using the pkCSM platform (**Table 6**). The results revealed high intestinal absorption rates for the newly designed compounds, ranging from 88.42% to 95.48%, with **Pred17** exhibiting the highest absorption rate suggesting excellent bioavailability.

Building on these absorption findings, the distribution properties of the compounds were analyzed, particularly their ability to penetrate the central nervous system (CNS) [35]. CNS penetration was evaluated using log PS values, where compounds with a LogPS value greater than −2 are considered effective in crossing the blood-brain barrier [36], while those with a LogPS less than −3 face significant challenges [37]. Among the tested compounds, **Pred16** (−2.179) showed minimal CNS penetration, indicating it is less likely to affect the CNS. In contrast, **5n** (−0.748) exhibited relatively higher permeability, suggesting better potential for CNS penetration.

In addition to distribution, the metabolic profiles of the compounds were assessed, focusing on their interactions with cytochrome P450 (CYP) enzymes. Among the 17 CYP families, 57 CYP genes have been identified in humans, with CYP enzymes playing a crucial role in the biotransformation of approximately 70–80% of clinically used drugs [38]. For this study, the analysis focused on CYP1A2, 2C19, 2D6, 3A4, and 2C9 enzymes. All proposed ligands were identified as substrates for CYP3A4 (**Table 6**), indicating their potential for metabolic processing by this key enzyme.

Following metabolism, the excretion properties of the compounds were evaluated through total clearance values (log ml/min/kg), which ranged from −0.105 (**Pred10**) to 0.55 (**5n**). Ligand **5n**, with the highest clearance value, may have a shorter half-life and could require more frequent dosing. Conversely, **Pred10**, with slower clearance, may provide prolonged therapeutic effects due to its extended presence in the system.

The toxicity profiles of the compounds were assessed to ensure their safety and efficacy. Each compound was evaluated using the AMES test, a widely recognized method for genotoxicity due to its simplicity, cost-effectiveness, and rapid results [39]. As shown in **Table 6**, most of the designed ligands were non-toxic. However, Pred1 exhibited some toxicity, raising concerns for its further development. These findings suggest that while most compounds are promising, certain modifications are needed to improve their non-toxicity, making them strong candidates for clinical evaluation.

### 3.6. Molecular docking study

Molecular docking is central to the design and screening of new bioactive molecules [40–42]. In this study, molecular docking of the three protein targets (CDK2, EGFR, and Tubulin) with all newly designed molecules was performed to identify

**Table 6. ADMET properties of selected molecules including the most active compound.**

| Ligands | Properties | | | | | | | | | | | |
|---|---|---|---|---|---|---|---|---|---|---|---|---|
| | Absorption | Distri-bution | Metabolism | | | | | | | | Excretion | Toxicity |
| | Human intestinal absorption | CNS | Cytochrome P450 (CYP450) | | | | | | | | Total Clearance | AMES toxicity |
| | | | | Substrate | | Inhibitor | | | | | | |
| | | | 2D6 | 3A4 | 1A2 | 2C19 | 2C9 | 2D6 | 3A4 | | | |
| | (%Absorbed) | log PS | Categorical (Yes/No) | | | | | | | | log ml/min/k | Categorical (Yes/No) |
| **Pred1** | 93.16 | −1.725 | Yes | Yes | No | Yes | Yes | No | No | 0.275 | Yes |
| **Pred9** | 90.917 | −1.787 | No | Yes | No | Yes | No | No | No | 0.198 | No |
| **Pred10** | 88.42 | −0.882 | Yes | Yes | Yes | Yes | Yes | Yes | No | −0.105 | No |
| **Pred15** | 92.796 | −1.787 | No | Yes | No | No | No | No | No | 0.299 | No |
| **Pred16** | 94.066 | −2.179 | No | Yes | Yes | Yes | Yes | No | Yes | 0.347 | No |
| **Pred17** | 95.486 | −0.810 | No | Yes | No | Yes | Yes | No | Yes | 0.268 | No |
| **5n** | 93.287 | −0.748 | Yes | Yes | Yes | Yes | Yes | No | Yes | 0.55 | No |

optimal binding modes that facilitate the inhibition of these targets. Virtual screening was employed to select molecules with the highest binding affinity scores. The docking procedures were validated as outlined in the methodology section, with Root Mean Square Deviation (RMSD) scores ranging from 1.08 to 1.83 Å. An RMSD value below 2.0 Å indicates a reliable predictive protocol for assessing protein-ligand interactions, confirming the docking protocols' appropriateness [43]. Additionally, parameters such as binding affinity, specific amino acid residues, and grid box dimensions were analyzed during validation. Lower binding affinity values signify stronger interactions between the ligand and the target [54]. Table 7 presents the binding affinity values for the most favorable interaction poses of all designed molecules with CDK2, EGFR, and Tubulin, using **molecule 5n** and FDA-approved drugs as controls for comparison. All the designed molecules demonstrate superior binding affinity compared to the reference drug and molecule 5n across all selected targets, with Pred9 and Pred10 exhibiting the better binding affinity among the six designed compounds. This suggests that the newly designed compounds could offer enhanced efficacy and stronger target interactions, highlighting their promising therapeutic potential.

Furthermore, the two-dimensional binding interaction of the compounds (Pred9, Pred10, Pred15, Pred16 and and Pred17), most active molecule 5n, and the reference drug for each protein target showed a similar interaction in the binding pocket of all the targets. This is due to several amino acids participating in the same interactions compared to the FDA drugs and molecule 5n.

In the context of CDK2, **Pred9** demonstrates strong binding stability through hydrogen bonds with Asn132 and Thr14, a pi-cation interaction with Lys129, and hydrophobic interactions with Ala31, Val18, and Lys33 (**Fig 5A**). Similarly, **Pred10** forms hydrogen bonds with Glu12 and Thr14, halogen interactions with Gln131, Asp145, and Asn132, and hydrophobic contacts with Ala31, Val64, and Phe80 (**Fig 5B**). For comparison, **Roscovitine**, a known CDK2 inhibitor, exhibits stabilizing interactions such as a pi-cation bond with Lys89, hydrogen bonds with Leu83, and hydrophobic contacts with Ile10 and Val18 (**Fig 5C**)

In the case of the Tubulin receptor, **Pred9** forms hydrogen bonds with Lys254 and Asn249, a pi-cation interaction with Lys254, and hydrophobic interactions with Leu248 and Cys241 (**Fig 6A**). Similarly, **Pred10** demonstrates significant binding through hydrogen bonds with Ser140 and Asn101, halogen interactions with Ser178 and Thr179, and hydrophobic contacts with Leu248 and Lys352 (**Fig 6B**). For comparison, **Colchicine**, a well-known Tubulin inhibitor, stabilizes through hydrogen bonds with Asn249 and Thr353, a pi-sigma interaction with Ser178, and hydrophobic contacts with Leu248 and Ala180 (**Fig 6C**)

For the EGFR receptor, **Pred9** exhibits stabilizing interactions, including a hydrogen bond and pi-anion interaction with Asp831, pi-sigma interactions with Val702, and hydrophobic contacts with Leu694 and Lys721 (**Fig 7A**). **Pred10** also

Table 7.  Docking score of the identified compounds against Cyclin-Dependent Kinase 2 (PDB ID: 2A4L), Tubulin (PDB ID: 1AS0), and Epidermal Growth Factor Receptor tyrosine kinase (PDB ID: 1M17) for anti-cancer activity.

| Compounds | Binding affinity (kcal/mol) | | |
| --- | --- | --- | --- |
| | CDK2 (2A4L) | Tubulin (1AS0) | EGFR (1M17) |
| Pred1 | −8.7 | −8.3 | −8.3 |
| Pred9 | −9.8 | −8.9 | −8.2 |
| Pred10 | −9.6 | −8.5 | −8.6 |
| Pred15 | −9 | −8.2 | −8.2 |
| Pred16 | −9.2 | −8.5 | −7.6 |
| Pred17 | −9 | −8.1 | −8.5 |
| 5n | −8.5 | −7.3 | −8.1 |
| Roscovitine | −8.2 | | |
| Colchicine | | −7.2 | |
| Erlotinib | | | −7.5 |

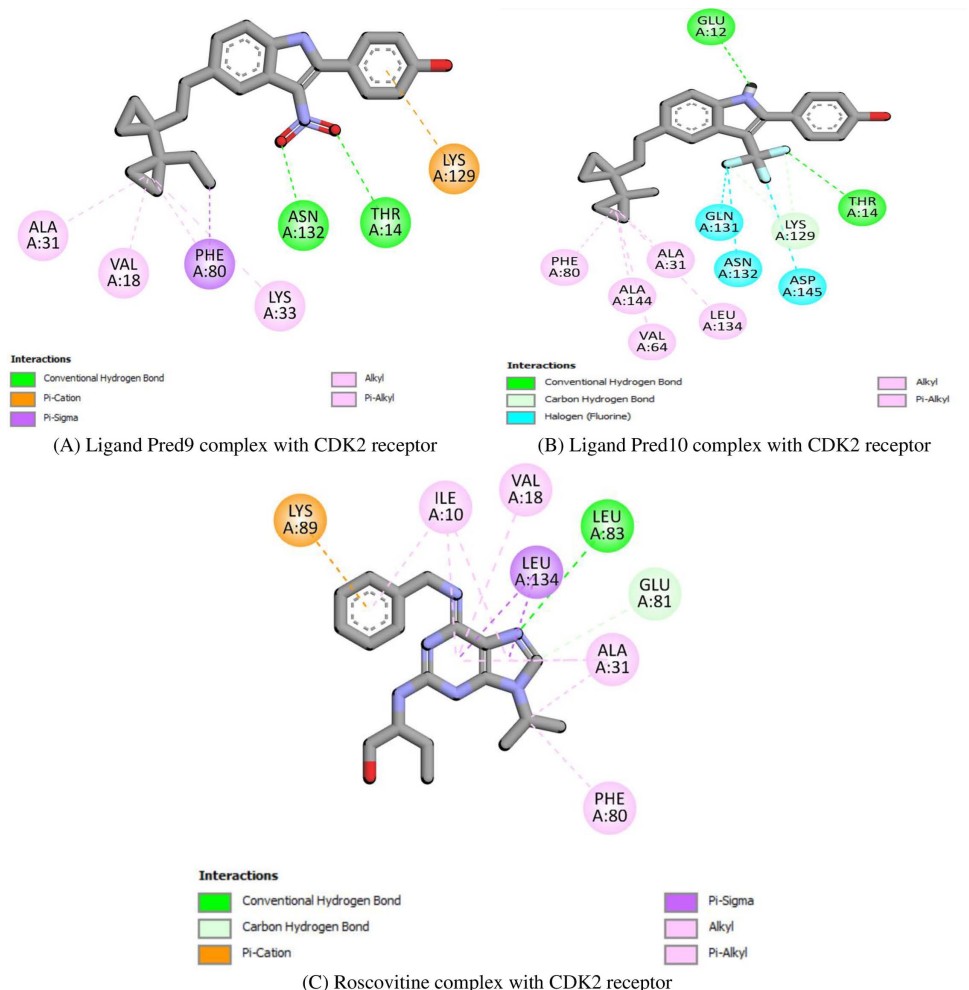

(A) Ligand Pred9 complex with CDK2 receptor

(B) Ligand Pred10 complex with CDK2 receptor

(C) Roscovitine complex with CDK2 receptor

**Fig 5. Docking simulation of the interaction between (A) Ligand Pred9, (B) Ligand Pred10 and (C) Roscovitine (reference drug) with CDK2 protein.**

strongly binds through a halogen bond with Asp831, hydrogen bonds with Glu738 and Met742, and hydrophobic interactions with Val702 and Leu820 (**Fig 7B**). For comparison, **Erlotinib**, an FDA-approved EGFR inhibitor, stabilizes through hydrogen bonds with Met769 and Cys773, a pi-cation interaction with Lys721, and hydrophobic contacts with Val702 and Leu764 (**Fig 7C**). These results emphasize the strong potential of the proposed compounds as multi-target drugs. Research indicates that multi-target compounds are particularly effective in treating complex diseases.

## 3.7. Molecular dynamics (MD)

Molecular dynamics (MD) simulations were conducted to evaluate the stability and dynamic interactions of protein-ligand complexes over time. These simulations modeled interatomic forces and generated trajectories to capture molecular fluctuations, providing detailed insights into interaction dynamics [44]. Specifically, MD simulations were performed for 100 ns on CDK2, EGFR, and Tubulin proteins complexed with Pred9, Pred10, and the most active compound. The resulting trajectories were analyzed to assess system stability and structural properties, including root mean square deviation (RMSD), root mean square fluctuation (RMSF), radius of gyration (Rg), and hydrogen bonding [23].

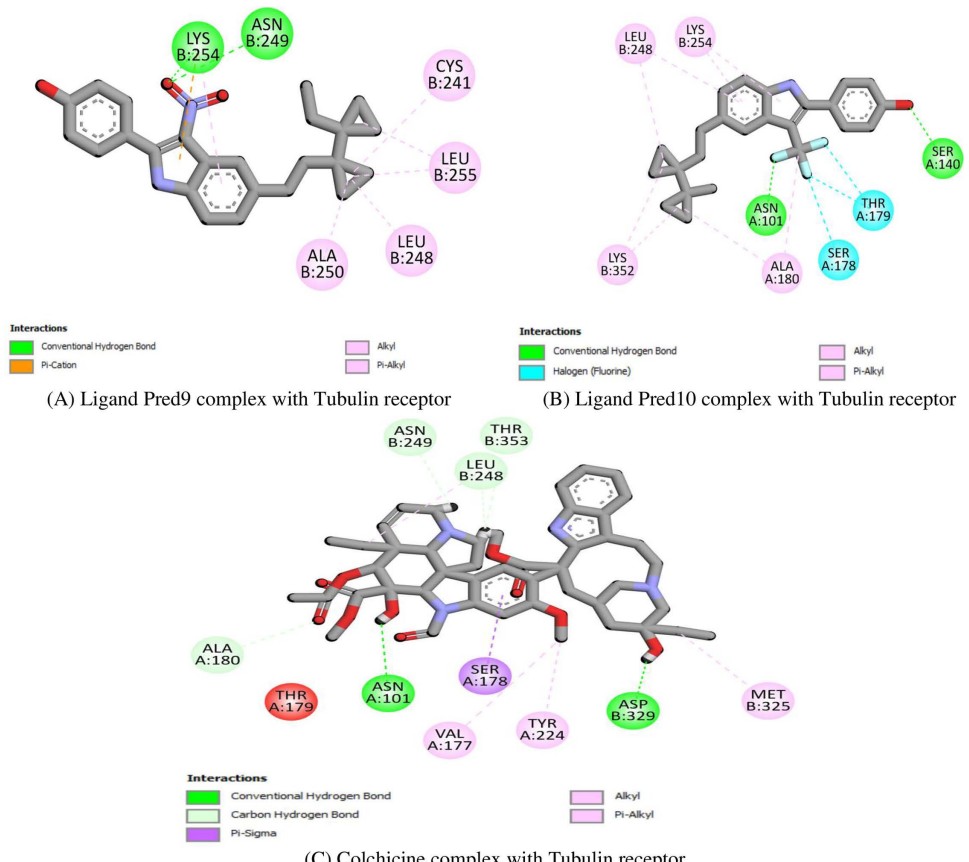

(A) Ligand Pred9 complex with Tubulin receptor
(B) Ligand Pred10 complex with Tubulin receptor
(C) Colchicine complex with Tubulin receptor

**Fig 6. Docking simulation of the interaction between (A) Ligand Pred9, (B) Ligand Pred10 and (C) Colchicine (reference drug) with Tubulin protein.**

**RMSD analysis.** RMSD measures the displacement of a protein's backbone atoms from their initial positions, providing insights into conformational stability. Lower RMSD values indicate greater stability, reflecting smaller deviations from the starting structure [45].

For the CDK2 complexes, the CDK2/Pred9 and CDK2/Pred10 complexes-maintained equilibrium with RMSD values ranging from 0.12 to 0.25 nm over the 100 ns MD simulation period (**Fig 8**). In contrast, the CDK2/Active molecule complex showed higher fluctuations. Similarly, the EGFR/Active molecule, EGFR/Pred9, and EGFR/Pred10 complexes exhibited comparable trajectories up to 80 ns. Beyond this point, the RMSD values for Pred9 and Pred10 stabilized and converged, demonstrating greater structural stability than the EGFR/Active molecule complex. For Tubulin, the Tubulin/Pred9 and Tubulin/Pred10 complexes achieved stability after 80 ns, while the Tubulin/Active molecule complex remained unstable throughout the simulation.

These results indicate that Pred9 and Pred10 exhibit the least RMSD fluctuations compared to the active compounds, suggesting that these designed compounds enhance target stability by minimizing protein backbone movement. This observation aligns with docking results, which showed better binding affinities for Pred9 and Pred10. However, RMSD alone cannot fully assess system stability, as it does not capture localized fluctuations. To address this limitation, RMSF analysis was performed to evaluate residue-specific variations throughout the trajectory.

**RMSF analysis.** RMSF is a crucial parameter in MD simulations used to assess protein flexibility and identify regions with significant structural variations [46]. By calculating RMSF values for each complex, it is possible to determine which

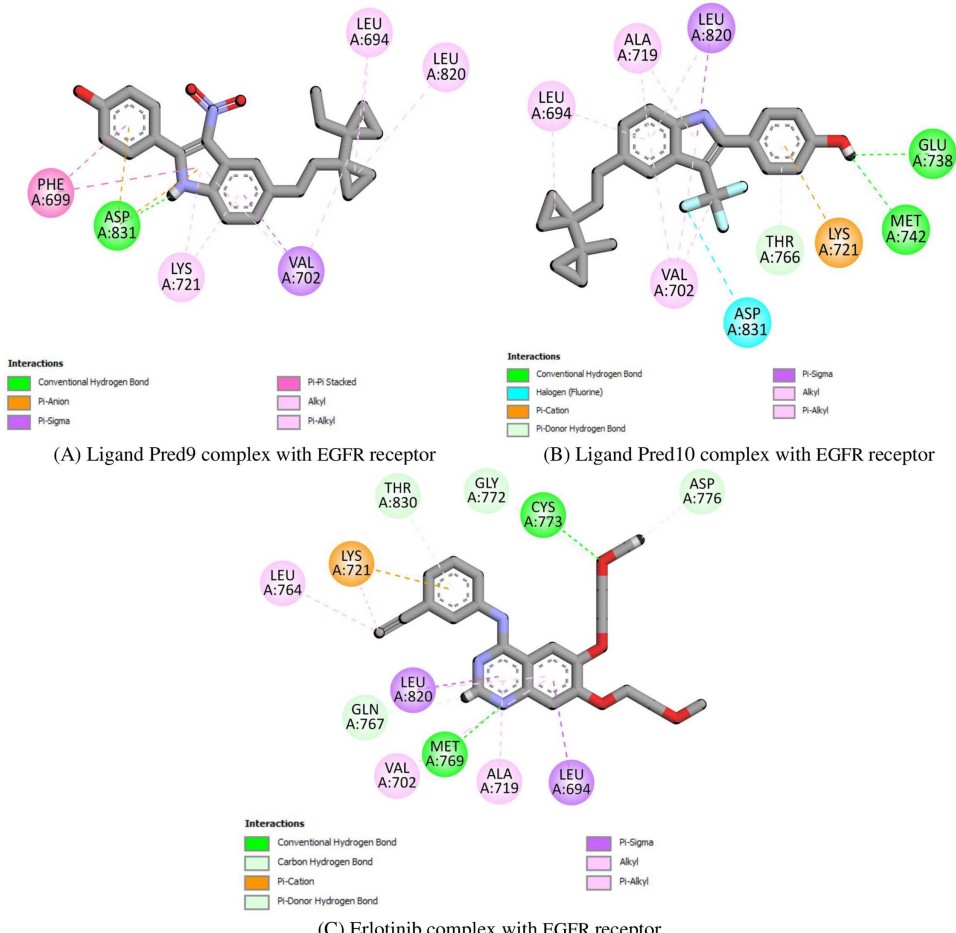

(A) Ligand Pred9 complex with EGFR receptor  (B) Ligand Pred10 complex with EGFR receptor

(C) Erlotinib complex with EGFR receptor

**Fig 7. Docking simulation of the interaction between (A) Ligand Pred9, (B) Ligand Pred10 and (C) Erlotinib (reference drug) with EGFR protein.**

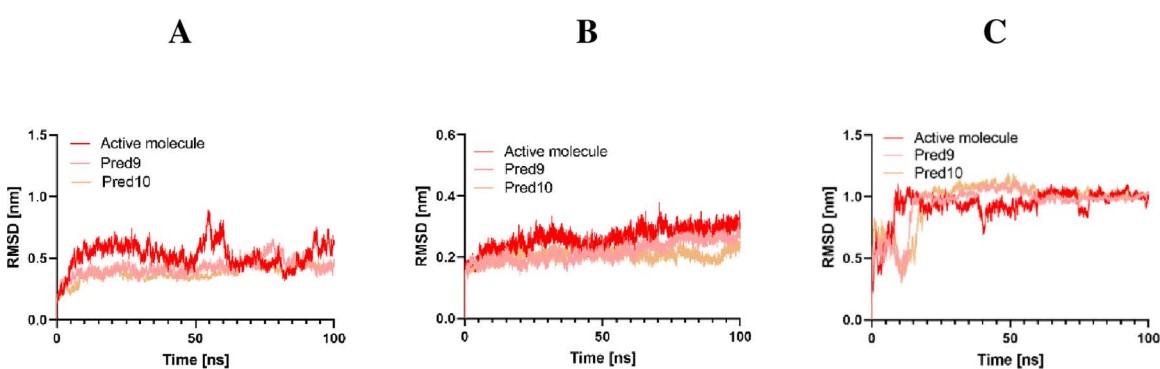

**Fig 8. RMSD Analysis of Active Molecules Pred9 and Pred10 in Complex with A: Tubulin, B: CDK2, and C: EGFR.**

residues experience the most pronounced fluctuations during the 100 ns MD simulation (**Fig 9**). Higher RMSF values indicate greater mobility of the protein's alpha carbon atoms, whereas lower values suggest more stable regions within the protein structure [47].

The designed compounds, Pred9 and Pred10, exhibited lower RMSF values compared to the active compounds when bound to EGFR, CDK2, and Tubulin. This reduced RMSF indicates that individual residues in these proteins experienced fewer atomic variations when complexed with Pred9 and Pred10, resulting in enhanced structural stability. The improved stability suggests that Pred9 and Pred10 form stronger and more consistent interactions with the protein targets, which may translate to more effective inhibition and therapeutic outcomes. These findings align with the RMSD results, where Pred9 and Pred10 also demonstrated lower deviations, indicating minimal backbone movement. The small fluctuations observed for Pred9 and Pred10 further highlight their ability to stabilize residual flexibility, resulting in fewer structural variations compared to the control compounds.

**Radius of gyration.** The radius of gyration (Rg) measures the compactness of a molecular structure by quantifying the average distance of its atoms from its center of mass [48]. **Fig 10** presents the radius of gyration for all the selected compounds. Compounds 9 and 10 exhibit a lower radius of gyration compared to the most active compounds when interacting with EGFR, CDK2, and Tubulin. This lower radius of gyration indicates a more compact and stable conformation of the protein-ligand complexes, suggesting that compounds 9 and 10 promote tighter binding and a more ordered arrangement within the binding sites. This enhanced compactness is consistent across all three protein targets, reflecting improved stabilization and potentially more effective inhibition

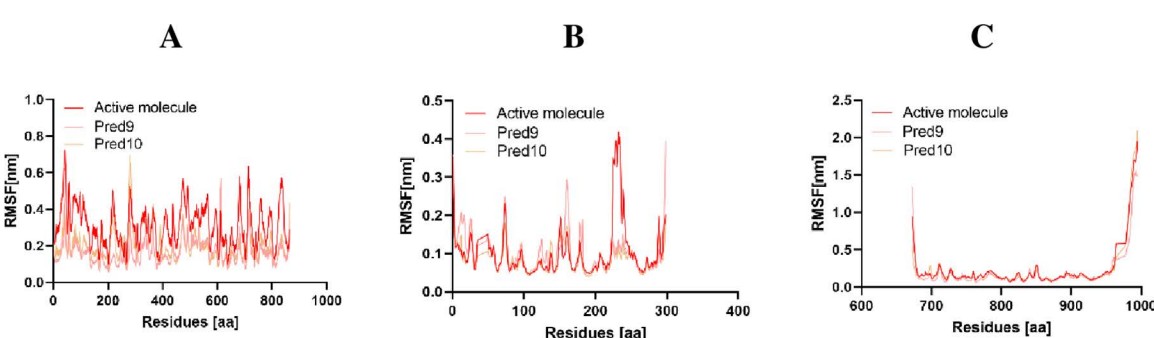

**Fig 9. RMSF Analysis of Active Molecule, Pred9, and Pred10 in Complex with A: Tubulin, B: CDK2, and C: EGFR.**

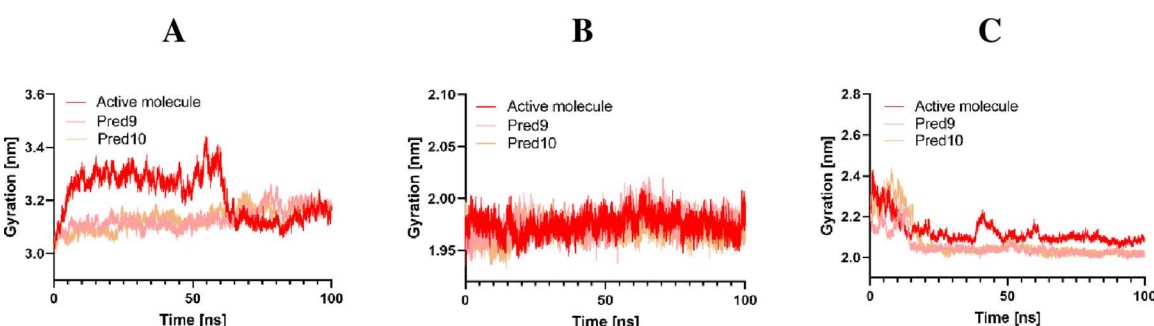

**Fig 10. Radius of gyration Analysis of Active Molecule, Pred9, and Pred10 in Complex with A: Tubulin, B: CDK2, and C: EGFR.**

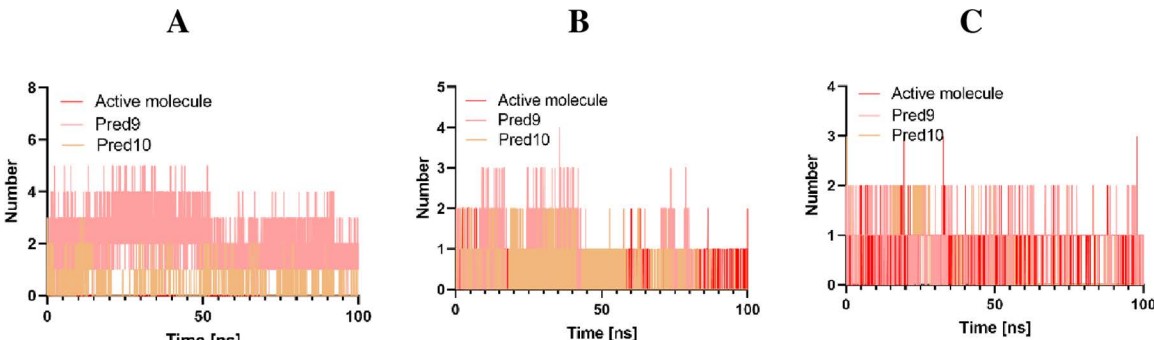

**Fig 11. H-bond Analysis of Active Molecule, Pred9, and Pred10 in Complex with A: Tubulin, B: CDK2, and C: EGFR.**

**Hbond analysis.** Hydrogen bonds play a critical role in molecular recognition by determining the directionality and specificity of interactions between proteins and chemicals [23]. To evaluate the stability of the docked complexes, we analyzed the hydrogen bonds formed by the designed compounds 9 and 10 (**Fig 11**), the most active compound and their target proteins (EGFR, CDK2, and Tubulin) during molecular dynamics (MD) simulations in a solvent environment.

Our calculations reveal that compounds 9 and 10 form a greater number of hydrogen bonds with these proteins compared to the most active compound, indicating more stable interactions. This observation is supported by the RMSD and RMSF plots shown in **Figs 8 and 9**, demonstrating that the designed compounds exhibit tighter binding than the most active compound. These results highlight the potential of compounds 9 and 10 as highly effective inhibitors of EGFR, CDK2, and Tubulin, crucial targets in cancer therapy. By enhancing the binding stability through increased hydrogen bond formation, these compounds could offer a more potent to disrupt tumor cell proliferation.

## 4. Conclusion

In this study, a 3D-QSAR analysis was performed using CoMFA and CoMSIA methods to develop a QSAR model correlating the biological activity of 2-Phenylindole derivatives against the MCF7 breast cancer cell line. The optimized CoMSIA/SEHDA model demonstrated strong reliability and predictive accuracy, as evidenced by validation metrics ($Q^2 = 0.814$, $R^2 = 0.967$, $R^2_{pred} = 0.722$). This model's contour maps designed six novel and enhanced anticancer inhibitors, with ADMET screening confirming their favorable pharmacokinetic profiles. Molecular docking studies revealed that the newly designed compounds exhibited better binding affinities, ranging from −7.2 to −9.8 kcal/mol, to key cancer-related targets (CDK2, EGFR, and Tubulin). Additionally, 100 ns molecular dynamics simulations confirmed the stability of the best-docked complexes within the binding pockets of these targets, highlighting their potential as promising candidates for anticancer drug development. However, it is important to emphasize that the clinical viability and safety of these compounds require further validation through in vitro and in vivo studies.

## Supporting information

**S1 Table. The selected targets and the coordinates of the grid box.**
(DOCX)

## Author contributions

**Conceptualization:** Mohamed Moussaoui, Khadijah M. Al-Zaydi, Soukayna Baammi.

**Data curation:** Mohamed Moussaoui, Khadijah M. Al-Zaydi, Soukayna Baammi.

**Formal analysis:** Khadijah M. Al-Zaydi.

**Investigation:** Khadijah M. Al-Zaydi.

**Methodology:** Mohamed Moussaoui, Khadijah M. Al-Zaydi, Soukayna Baammi.

**Project administration:** Khadijah M. Al-Zaydi.

**Resources:** Khadijah M. Al-Zaydi.

**Software:** Mohamed Moussaoui, Khadijah M. Al-Zaydi, Soukayna Baammi.

**Supervision:** Khadijah M. Al-Zaydi.

**Validation:** Mohamed Moussaoui, Khadijah M. Al-Zaydi, Soukayna Baammi.

**Visualization:** Mohamed Moussaoui, Khadijah M. Al-Zaydi, Soukayna Baammi.

**Writing – original draft:** Mohamed Moussaoui, Khadijah M. Al-Zaydi, Soukayna Baammi.

**Writing – review & editing:** Mohamed Moussaoui, Khadijah M. Al-Zaydi, Soukayna Baammi.

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
