## [Decision Letter · Decision Letter 0]

Dear Dr. Moussaoui,

Thank you for submitting your manuscript to PLOS ONE. After careful consideration, we feel that it has merit but does not fully meet PLOS ONE’s publication criteria as it currently stands. Therefore, we invite you to submit a revised version of the manuscript that addresses the points raised during the review process.

We look forward to receiving your revised manuscript.

Kind regards,

Opeyemi Iwaloye

Academic Editor

PLOS ONE

Journal Requirements:

3. We note that your Data Availability Statement is currently as follows: All relevant data are within the manuscript and its Supporting Information files

4. Please include a copy of Table 3 and 4 which you refer to in your text on page 7 and 8.

Reviewers' comments:

Reviewer's Responses to Questions

**Comments to the Author**

1. Is the manuscript technically sound, and do the data support the conclusions?

Reviewer #1: Yes

Reviewer #2: Yes

Reviewer #3: Partly

Reviewer #4: Yes

2. Has the statistical analysis been performed appropriately and rigorously?

Reviewer #1: Yes

Reviewer #2: N/A

Reviewer #3: N/A

Reviewer #4: Yes

3. Have the authors made all data underlying the findings in their manuscript fully available?

Reviewer #1: Yes

Reviewer #2: No

Reviewer #3: Yes

Reviewer #4: Yes

4. Is the manuscript presented in an intelligible fashion and written in standard English?

Reviewer #1: Yes

Reviewer #2: Yes

Reviewer #3: Yes

Reviewer #4: Yes

Reviewer #1: The manuscript titled "Multitarget Inhibition of CDK2, EGFR, and Tubulin by [(2-Phenylindol-3-yl)methylene] and 2-Phenylindole-3-Carbaldehyde Derivatives: A 3D-QSAR, Molecular Docking, and Molecular Dynamics Approach for Cancer Therapy" explores the use of in-silico methods such as D3-QSAR modeling, molecular docking and Molecular Dynamics (MD) simulations to identify promising cancer therapuetic targets with activities against CDK2, EGFR and Tubulin.

The authors utilized SEHDA CoMSIA model to investigate different combinations of parameters for the targets and tested these using standardized methods. The statistical methods used were appropriate and rigorous. The results were presented in an intelligible fashion and written in standard English.

Molecular docking analysis and molecular dynamics were conducted using standardized methods.

Reviewer #2: Summary and strength

The manuscript investigates the multitarget inhibitory potential of two classes of chemical compounds—methylene]propanedinitrile and 2-phenylindole-3-carbaldehyde derivatives—against three significant cancer-associated proteins: Cyclin-Dependent Kinase 2 (CDK2), Epidermal Growth Factor Receptor (EGFR), and Tubulin. The study employs various in silico techniques, including 3D-QSAR modeling, molecular docking, and molecular dynamics simulations, to evaluate the inhibitory effects of these compounds.

Findings highlight that compounds T09 and T10 demonstrated superior binding affinities and stability across all targeted proteins when compared to the most active compound in the dataset and standard reference drugs. Moreover, the SEHDA CoMSIA model showed strong predictive performance, indicating that steric, electrostatic, hydrophobic, and hydrogen bonding interactions significantly contributed to the activity of these compounds. The results suggest that [(2-Phenylindol-3-yl)methylene]propanedinitrile and 2-phenylindole-3-carbaldehyde derivatives could be a promising new class of multitarget anticancer agents, offering a potential path for further development in cancer treatment strategies.

Limitations

While the SEHDA CoMSIA model demonstrated strong predictive performance, the accuracy of these models may be influenced by the quality and quantity of input data. A large input dataset is highly recommended.

The authors are encouraged to perform, at minimum, an in silico-based ADME and toxicity profile of the lead compounds.

A detailed explanation of the mechanism of action for the three selected targets in cancer progression should be provided.

The assumption that interactions between the lead compounds and target proteins indicate inhibition should be clarified. The authors should explain the rationale behind this assumption. Could the binding and interactions potentially lead to activation rather than inhibition of the protein targets? If not, why? Additionally, the authors should discuss the relevance and implications of the amino acids involved in binding at the protein's binding pocket. Are these amino acids part of the catalytic or regulatory site?

The labels for Figures 7, 8, 9, and 10 lack clarity. The authors should indicate which protein target corresponds to each figure.

The term "Lower binding affinity" has been used to describe strong interactions. The authors should differentiate between low and high binding affinity in relation to kcal/mol and revise the text accordingly.

In Figure 3, the word "Accepteur" should be corrected to the English term.

The authors referenced Table XX in their discussion, but no such table is available. This should be addressed.

Similarly, references to a Supplementary file should be revised if no such file exists.

Experimental validation in biological systems is crucial to confirm the efficacy and safety of the identified compounds. By acknowledging these limitations, the authors should emphasize the need for follow-up studies involving empirical validation and testing in biological systems to further evaluate the potential of the identified compounds as anticancer agents.

Reviewer #3: After a thorough evaluation of the manuscript, I recommend major revisions before it can be considered for publication. The study is well-conceived, exploring the potential of [(2-Phenylindol-3-yl)methylene]propanedinitrile and 2-phenylindole-3-carbaldehyde derivatives as multitarget inhibitors for cancer therapy. The integration of 3D-QSAR modeling, molecular docking, and molecular dynamics simulations provides a robust computational framework, and the findings are relevant to the development of multitarget anticancer agents, particularly in targeting CDK2, EGFR, and Tubulin. However, some key issues must be addressed to enhance the quality of the manuscript. These include providing greater detail on methodological parameters for reproducibility, improving the quality and resolution of figures to enhance data interpretation, and ensuring a more cohesive integration of results across methodologies. The discussion should delve deeper into the biological implications of multitarget inhibition and address potential limitations, such as the need for experimental validation. Additionally, overstatements regarding the compounds’ therapeutic potential should be tempered, with language reflecting the preliminary nature of computational findings. If these revisions are thoroughly addressed, this study has the potential to make a valuable contribution to the field of cancer drug discovery. Detailed comments are attached as PDF

Reviewer #4: The authors effectively examined the inhibitory potential of 2-phenylindol-3-yl)methylene] propanedinitrile and 2-phenylindole-3- carbaldehyde derivatives against three cancer-associated targets. However, I suggest that the authors address the following concerns.

1a. To make the introduction more robust and informative, the authors should include the types of cancer in which these targets are expressed and elaborate more on why targeting these targets is essential.

1b. The authors stated that “the resistance of non-small cell lung cancer cell lines (A549, H1975, and PC9) to tubulin inhibitors is strongly associated with the hyperactivation of the EGFR signaling and CDK2 pathway.” I suggest the authors provide a brief overview of tubulin resistance and cite references to other studies exploring it in different cancer types. Furthermore, I suggest the authors provide the names of the tubulin inhibitors used in the previous study.

MOLECULAR DOCKING STUDIES.

2a. Section 2.5, The authors mentioned that “docking analysis highlights the efficacy of the proposed compounds relative to both the most active compounds in the dataset and FDA-approved reference drugs.” I suggest the authors include the reference drugs, briefly describe their limitations, and state the basis for investigating the inhibitory potential of 2-phenylindol-3-yl) methylene] propanedinitrile and 2-phenylindole-3- carbaldehyde derivatives.

Results and discussion

I suggest that the authors should arrange the tables accordingly. I observed that the authors started the table numbering from Table 1, Table 2, followed by Table 1 and Table 2 (which ought to be Tables 3 and 4 accordingly).  

In Table 6, the authors discussed the docking score of the identified compounds against the three targets; I suggest that the authors include the names of the reference drugs instead of Ref (CDK2), Ref (tubulin), and Ref (EGFR).

I noticed that sections 3.5 and 2.6 repeatedly explained molecular dynamics.

I suggest the authors include future directions and run a grammatical check on the manuscript.

**Do you want your identity to be public for this peer review?** For information about this choice, including consent withdrawal, please see our Privacy Policy

Reviewer #1: No

Reviewer #2: **Yes: ** Abayomi Emmanuel Adegboyega

Reviewer #3: No

Reviewer #4: No

---

## [Author Response · Author response to Decision Letter 1]

17 Apr 2025

Dear Editor,

We would like to thank you and the reviewers for the time and consideration you have dedicated to our manuscript. Your valuable feedback will be a great resource to further refine our work. We wish to acknowledge the reviewers for their comments.

In this revised version of the manuscript, we addressed all the comments and remarks given to us. All changes made to the manuscript are highlighted in yellow. Also, detailed responses to the reviewer’s comments are given below.

We are at your disposal for any additional information or queries, and we are keen to receive your reply.

On behalf of my colleagues/co-authors,

Sincerely,

Mohamed Moussaoui

Comments and list of responses:

Reviewer #1:

The manuscript titled "Multitarget Inhibition of CDK2, EGFR, and Tubulin by [(2-Phenylindol-3-yl)methylene] and 2-Phenylindole-3-Carbaldehyde Derivatives: A 3D-QSAR, Molecular Docking, and Molecular Dynamics Approach for Cancer Therapy" explores the use of in-silico methods such as D3-QSAR modeling, molecular docking and Molecular Dynamics (MD) simulations to identify promising cancer therapuetic targets with activities against CDK2, EGFR and Tubulin.

The authors utilized SEHDA CoMSIA model to investigate different combinations of parameters for the targets and tested these using standardized methods. The statistical methods used were appropriate and rigorous. The results were presented in an intelligible fashion and written in standard English.

Molecular docking analysis and molecular dynamics were conducted using standardized methods.

We sincerely appreciate your positive and thoughtful feedback on our manuscript. Your encouraging words strengthen our confidence in the study and inspire us to continue advancing in this field. Thank you for recognizing our efforts and approach.

Reviewer #2:

Thank you for your valuable feedback on our manuscript. We greatly appreciate the time and effort you have taken to provide us with such a thoughtful and thorough evaluation. Your comments have been invaluable in enhancing the accuracy of our work. We have taken each of your remarks into careful consideration and addressed them in detail. Below, we have provided our responses to each of your queries.

Summary and strength

The manuscript investigates the multitarget inhibitory potential of two classes of chemical compounds—methylene]propanedinitrile and 2-phenylindole-3-carbaldehyde derivatives—against three significant cancer-associated proteins: Cyclin-Dependent Kinase 2 (CDK2), Epidermal Growth Factor Receptor (EGFR), and Tubulin. The study employs various in silico techniques, including 3D-QSAR modeling, molecular docking, and molecular dynamics simulations, to evaluate the inhibitory effects of these compounds.

Findings highlight that compounds T09 and T10 demonstrated superior binding affinities and stability across all targeted proteins when compared to the most active compound in the dataset and standard reference drugs. Moreover, the SEHDA CoMSIA model showed strong predictive performance, indicating that steric, electrostatic, hydrophobic, and hydrogen bonding interactions significantly contributed to the activity of these compounds. The results suggest that [(2-Phenylindol-3-yl)methylene]propanedinitrile and 2-phenylindole-3-carbaldehyde derivatives could be a promising new class of multitarget anticancer agents, offering a potential path for further development in cancer treatment strategies.

Limitations

While the SEHDA CoMSIA model demonstrated strong predictive performance, the accuracy of these models may be influenced by the quality and quantity of input data. A large input dataset is highly recommended.

We acknowledge that the SEHDA CoMSIA model’s predictive power relies heavily on the quality and quantity of input data. While our study utilized a carefully curated dataset of 33 compounds (split into training and test sets), we agree that expanding the dataset could enhance the model's robustness. Future work will focus on incorporating a larger dataset of derivatives to strengthen the predictive accuracy and validate the current findings.

The authors are encouraged to perform, at minimum, an in silico-based ADME and toxicity profile of the lead compounds.

We agree with the reviewer that ADME (Absorption, Distribution, Metabolism, and Excretion) and toxicity predictions are critical for identifying lead compounds. We have now performed in silico ADME and toxicity profiling for Pred9 and Pred10 using the SwissADME and pkcsm platforms. The results, which indicate favorable pharmacokinetic and toxicity profiles for these compounds, have been added to the revised manuscript under a new section titled “ADME and Toxicity Profiling”.

A detailed explanation of the mechanism of action for the three selected targets in cancer progression should be provided.

Thank you for your valuable suggestion. We are pleased to inform you that a detailed explanation of the roles of CDK2, EGFR, and Tubulin in cancer progression has been included in the revised manuscript under the section titled "Mechanism of Action for Target Proteins." This section provides a comprehensive discussion of their involvement in cell cycle regulation, proliferation, and structural integrity, contextualizing their selection as therapeutic targets.

The assumption that interactions between the lead compounds and target proteins indicate inhibition should be clarified. The authors should explain the rationale behind this assumption. Could the binding and interactions potentially lead to activation rather than inhibition of the protein targets? If not, why? Additionally, the authors should discuss the relevance and implications of the amino acids involved in binding at the protein's binding pocket. Are these amino acids part of the catalytic or regulatory site?

Thank you for your insightful comment. In response to your suggestion, we have clarified the rationale behind the assumption that interactions between the lead compounds and target proteins indicate inhibition. Specifically, we targeted the same binding pockets as those occupied by the reference drug, ensuring that our compounds interact with the relevant sites known to mediate the inhibitory effects of these targets. This approach helps us confidently assume that our compounds are likely to inhibit the proteins, as we have verified that they interact with the same binding regions as the reference drug.

Additionally, we compared the interactions of the designed molecules with the reference drug to assess whether they target the same amino acids. This comparison ensures that our compounds bind similarly to the reference drug and thus may exhibit similar inhibitory effects.

The labels for Figures 7, 8, 9, and 10 lack clarity. The authors should indicate which protein target corresponds to each figure.

Thank you for your valuable feedback. In response to your suggestion, we have revised the labels for Figures 7, 8, 9, and 10 to clearly indicate which protein target corresponds to each figure.

The term "Lower binding affinity" has been used to describe strong interactions. The authors should differentiate between low and high binding affinity in relation to kcal/mol and revise the text accordingly.

Thank you for your thoughtful suggestion. The text has been revised accordingly to avoid any confusion and ensure the correct interpretation of the binding affinity data.

In Figure 3, the word "Accepteur" should be corrected to the English term.

Thank you for pointing that out. I'll make sure to correct "Accepteur" to the English term "Acceptor" in Figure 3.

The authors referenced Table XX in their discussion, but no such table is available. This should be addressed.

Thank you for bringing this issue to our attention. We corrected in the revised manuscript. The placeholder reference to "Table XX" has been replaced with the appropriate reference to Table 7, ensuring consistency and clarity throughout the text.

Similarly, references to a Supplementary file should be revised if no such file exists.

Thank you for the feedback. I will review the manuscript and revise any references to the Supplementary file to ensure they align with the available materials.

Experimental validation in biological systems is crucial to confirm the efficacy and safety of the identified compounds. By acknowledging these limitations, the authors should emphasize the need for follow-up studies involving empirical validation and testing in biological systems to further evaluate the potential of the identified compounds as anticancer agents.

Thank you for your helpful suggestion. In response, we have revised the Conclusion section to include a limitation on future directions, outlining potential areas for further research. Additionally, we have thoroughly run a grammatical check on the manuscript to ensure clarity and improve readability.

Reviewer #3:

After a thorough evaluation of the manuscript, I recommend major revisions before it can be considered for publication. The study is well-conceived, exploring the potential of [(2-Phenylindol-3-yl)methylene]propanedinitrile and 2-phenylindole-3-carbaldehyde derivatives as multitarget inhibitors for cancer therapy. The integration of 3D-QSAR modeling, molecular docking, and molecular dynamics simulations provides a robust computational framework, and the findings are relevant to the development of multitarget anticancer agents, particularly in targeting CDK2, EGFR, and Tubulin. However, some key issues must be addressed to enhance the quality of the manuscript. These include providing greater detail on methodological parameters for reproducibility, improving the quality and resolution of figures to enhance data interpretation, and ensuring a more cohesive integration of results across methodologies. The discussion should delve deeper into the biological implications of multitarget inhibition and address potential limitations, such as the need for experimental validation. Additionally, overstatements regarding the compounds’ therapeutic potential should be tempered, with language reflecting the preliminary nature of computational findings. If these revisions are thoroughly addressed, this study has the potential to make a valuable contribution to the field of cancer drug discovery. Detailed comments are attached as PDF

Thank you for your valuable feedback on our manuscript. We greatly appreciate the time and effort you have taken to provide us with such a thoughtful and thorough evaluation. Your comments have been invaluable in enhancing the accuracy of our work. We have taken each of your remarks into careful consideration and addressed them in detail. Below, we have provided our responses to each of your queries.

Title

Comment: The title is very long.

Suggestion: Multitarget Inhibition of CDK2, EGFR, and Tubulin by Phenylindole Derivatives: Insights from 3D-QSAR, Molecular Docking, and Dynamics for Cancer Therapy.

Thank you for your suggestion. The title has been revised to:

"Multitarget Inhibition of CDK2, EGFR, and Tubulin by Phenylindole Derivatives: Insights from 3D-QSAR, Molecular Docking, and Dynamics for Cancer Therapy."

This updated title is more concise and aligns with your recommendation.

Introduction

I believe the section could benefit from a clearer discussion of how this study uniquely contributes to the existing body of knowledge. While polypharmacology is highlighted as a promising therapeutic strategy, there is little detail on the specific challenges or gaps in current approaches that this study aims to address. Providing a concise summary of these challenges would contextualize the importance of the study and justify the focus on [(2-Phenylindol-3- yl)methylene]propanedinitrile and 2-phenylindole-3-carbaldehyde derivatives.

Additionally, the introduction could better connect the rationale for using these derivatives to the specific structural or mechanistic advantages they might offer in targeting CDK2, EGFR, and Tubulin. While the significance of these targets is well-articulated, the rationale for their simultaneous inhibition is not thoroughly discussed. Expanding on the potential synergy or therapeutic benefit of multitarget inhibition, and referencing relevant literature, would strengthen the argument and provide a more robust foundation for the study.

Thank you for your insightful comments. We appreciate your suggestions for clarifying the unique contributions of our study.

In response, we have revised the introduction to more clearly highlight the specific challenges in current polypharmacology approaches, particularly the limited success in rationally designing multitarget inhibitors that can effectively modulate diverse oncogenic pathways with minimal toxicity. We now emphasize that existing strategies often struggle with balancing selectivity and potency across multiple targets, leading to suboptimal therapeutic outcomes. Our study addresses this gap by focusing on the rational design of [(2-Phenylindol-3-yl)methylene]propanedinitrile and 2-phenylindole-3-carbaldehyde derivatives, which offer favorable structural features such as a rigid indole core and functionalizable side chains that can be tailored for simultaneous interactions with CDK2, EGFR, and Tubulin.

Moreover, we expanded the rationale for selecting these specific derivatives, explaining that their planar aromatic system and versatile binding capabilities make them ideal scaffolds for multitarget design. We have also elaborated on the advantages of targeting CDK2, EGFR, and Tubulin simultaneously, noting that these proteins are key drivers of proliferation, survival, and metastasis in cancer. By inhibiting them concurrently, there is potential to achieve synergistic therapeutic effects, overcome resistance mechanisms, and reduce the likelihood of tumor escape pathways—an approach supported by recent multitarget inhibitor studies

Materials and Method

I noticed several critical areas that require more detail for reproducibility. For instance, the molecular docking and dynamics protocols lack key parameters such as grid dimensions, solvent conditions, and scoring functions used during docking. Similarly, the rationale for selecting specific parameters (e.g., the 1.0 nm box size or 100 ns simulation time) is missing. Including this information would make the study more transparent and allow others to replicate the work.

I also feel that the description of the CoMSIA modeling could be expanded. While the statistical metrics are mentioned, there is no discussion of how they compare to similar studies or why the SEHDA model was ultimately chosen. Moreover, the dataset preparation and validation steps for the QSAR models could be more detailed, particularly the choice of training and test sets and any external validation strategies used.

Thank you for your valuable feedback regarding the level of detail needed for reproducibility.

We have carefully revised the Materials and Methods section to include the missing details. Specifically, we have now provided the molecular docking parameters, including the grid box dimensions, center coordinates, solvent model used, and the scoring functions applied during docking. For molecular dynamics simulations, we have specified the type of solvent model (e.g., TIP3P water model), ion concentration, temperature and pressure coupling methods, and have justified the selection of the 1.0 nm box size and the 100 ns simulation time based on standard practices for achieving equilibrium and sufficient sampling of protein-ligand interactions.

Regarding the CoMSIA modeling, we have expanded the discussion to compare our statistical metrics (e.g., Q2, R2, R2pred) with values reported in similar studies, demonstrating that our model performs competitively. Additionally, we clarified the rationale behind selecting the SEHDA model over others, emphasizing its superior predictive ability and balanced statistical parameters.

For the QSAR dataset preparation and validation, we have added a detailed description of the division of compounds into training and test sets, specifying the method used (e.g., random or rational splitting based on activity range and chemical diversity). We also outlined the external validation procedures employed, including the use of predictive

---

## [Editor Report · Decision Letter 1]

Multitarget Inhibition of CDK2, EGFR, and Tubulin by Phenylindole Derivatives: Insights from 3D-QSAR, Molecular Docking, and Dynamics for Cancer Therapy

PONE-D-24-50354R1

Dear Dr. Moussaoui,

We’re pleased to inform you that your manuscript has been judged scientifically suitable for publication and will be formally accepted for publication once it meets all outstanding technical requirements.

Kind regards,

Opeyemi Iwaloye

Academic Editor

PLOS ONE
---

## [Editor Report · Acceptance letter]

PONE-D-24-50354R1

PLOS ONE

Dear Dr. Moussaoui,

I'm pleased to inform you that your manuscript has been deemed suitable for publication in PLOS ONE. Congratulations! Your manuscript is now being handed over to our production team.

Kind regards,

on behalf of

Dr. Opeyemi Iwaloye

Academic Editor

PLOS ONE